# An Investigation of the Wishes, Needs, Opportunities and Challenges of Accessing Meaningful Activities for People Living with Mild to Moderate Dementia

**DOI:** 10.3390/ijerph20075358

**Published:** 2023-03-31

**Authors:** Isabelle Tournier, Laura Orton, Tom Dening, Anya Ahmed, Vjera Holthoff-Detto, Kristina Niedderer

**Affiliations:** 1Department of Design, Manchester School of Art, Manchester Metropolitan University, Manchester M15 6BR, UK; 2Department of Psychology, Laboratoire Cliniques Pathologique et Interculturelle, Université Toulouse 2 Jean Jaurès, 31058 Toulouse, France; 3Mental Health & Clinical Neurosciences, School of Medicine, University of Nottingham, Nottingham NG7 2UH, UK; 4Department of Social Care and Social Work, Manchester Metropolitan University, Manchester M15 6GX, UK; 5Alexianer Krankenhaus Hedwigshöhe, 12526 Berlin, Germany; 6Department of Psychiatry and Psychotherapy, Faculty of Medicine Carl Gustav Carus, University of Technology Dresden, 01307 Dresden, Germany

**Keywords:** mild dementia, moderate dementia, meaningful activities, social participation, empowerment, needs, preferences, barriers, facilitators, services

## Abstract

Many people are keen to be actively involved in social life and activities, but even at an early stage, dementia can have a negative impact on social participation and access to leisure activities. As part of the IDoService project, this study has investigated people’s needs and wishes, barriers and facilitators to identify opportunities for improving access to meaningful activities. Individual and focus group interviews were conducted with 5 people living with mild to moderate dementia, 2 familial and 2 professional care partners, as well as 12 people working in the field of dementia and/or community activities. Thematic analysis has highlighted the benefits of participating in meaningful activities, such as empowerment and pride, social contacts, and feeling useful to others. A number of barriers to participation relating to individual and environmental factors were reported. Even where participants praised dementia-friendly activities and facilities, they advocated activities inclusive for all and mentioned that some people might be reluctant to participate in dementia-labelled activities because they may not be suitable for their needs. These results indicate the need for developing tailored opportunities for people with mild to moderate dementia and provide valuable insights for researchers, service providers, policymakers and charities wanting to improve access.

## 1. Introduction

Supporting people with dementia in their daily living is one of the key European priorities due to the high prevalence of dementia in old age and the exponential increase of cases—from approximately 55 million worldwide in 2019 to a predicted 132 million by 2050 [1]. Given the lack of a medical cure, the implementation and development of psychosocial interventions supporting the wellbeing of people with dementia is a key aspect [2]. Promoting active participation in purposeful and social activities benefits people with dementia and increases their quality of life [3,4]. However, there is, as yet, little research about the access for, and social inclusion of, people with mild to moderate dementia to social activities and how to promote it. The challenge in supporting people with mild dementia is the need to enable participation in individually tailored activities to cater to their interests and needs and to avoid stigmatisation.

The aim of the IDoService project is to co-develop, implement and evaluate such a tailored service to increase the sociocultural participation and empowerment of people living with mild to moderate stages of dementia in Greater Manchester, UK. In order to inform the service design, individual interviews and focus groups were conducted to collect insights about the needs and preferences of people with mild to moderate dementia regarding their involvement in sociocultural and leisure activities. The investigation focused on barriers and facilitators as well as existing strategies and future opportunities for promoting participation in meaningful activities to ascertain the baseline criteria for the service. 

### 1.1. Participation in Meaningful Activities for People Living with Dementia

Meaningful activities comprise physical, social and leisure activities that are tailored to a person’s needs and preferences. This includes daily living activities (e.g., dressing and eating) as well as leisure activities, such as reading, singing, gardening or arts and crafts [5]. According to Tierney and Beattie [6], there are five attributes that make an activity meaningful for a person: being enjoyable, suited to the individual’s skills, abilities and preferences, related to personally relevant goals, engaging, and related to an aspect of identity. Dementia is a progressive condition that gradually challenges people’s ability to perform activities due to cognitive and behavioural changes as well as the ability of family members and care partners to effectively support active engagement in leisure activities [7]. This is concerning because participation in meaningful activities provides emotional, creative, and intellectual stimulation [8,9] . Being active and engaging in leisure and social activities as much as possible is a major imperative for people with dementia and an acknowledged means of empowerment [10]. This can lead to feelings of pleasure and enjoyment, a sense of connection and belonging, as well as a sense of autonomy and personal identity [11,12] . However, investigations regarding the impact of mild symptoms of dementia on social participation and how to facilitate access to activities are still scarce. First investigations suggest benefits for quality of life, delay functional decline [11,13] and improvements regarding stress [14]. 

### 1.2. Needs for and Barriers to Active Participation in Meaningful Activities

Participation in activities is often reported as an unmet need by people living with dementia [15]. People are looking for more self-determination and opportunities to conduct activities on their own, to continue to learn new things and be creative, as well as feel useful to others, have social contacts and give meaning to life [16]. The capacity of people with dementia to be involved in daily activities depends on the match between the person’s and the environment’s adaptive capacity [17]. As a result, people might not be able to engage with an activity in its original form but be able to pursue it if it is modified and personalised to suit their skills [18]. Without adaptation, participation in leisure opportunities can become limited to home-based or dementia-specific services [8]. Barriers to participation can include a lack or poor accessibility of transport and venues, worrying about getting lost or confused or potentially feeling overwhelmed in crowded situations [8,11]. Social isolation and lack of community support, apathy, embarrassment about symptoms of dementia and a lack of understanding or tolerance by staff and others, as well as electronic payment methods in shops or self-service checkouts, can also be barriers to social participation [19] . 

### 1.3. Local Community Services as Facilitators for Supporting Participation

Barriers to participation can be at least partially overcome by the provision of appropriate community services offering accessible leisure and social activities. Local clubs and organisations providing a supportive environment are also beneficial for engaging in social and leisure activities [20,21] . The dementia-friendliness of communities can be increased by offering environmental support that encompasses age-friendly features [22] in order to celebrate the “capabilities of people with dementia and view them as valuable and vital members of the towns, cities, villages and countries in which they reside” [23]. However, available activities and services are often not adapted to the needs of people in the earlier stages of dementia or with early-onset dementia, or the dedicated activities are perceived as too dull [20]. The latter are also often too focused on medical needs and long-term planning, neglecting to involve people actively and empower them by focusing on their capabilities [24]. 

There is increasing recognition of this lack of suitable or accessible services, and a number of innovative approaches to facilitate engagement in meaningful activities are emerging, such as the use of digital technologies for creating an individualised offer for people with dementia living in the community [25] or care-home settings [26]. However, digital applications cannot always solve the lack of suitable provision and accessibility, and research is required to understand the nature and level of the provision and support required.

Service design can help to increase the dementia-friendliness of services by offering a holistic, co-creative and person-centred approach to improve the quality-of-service provision [27]. This holistic view includes the processes, components, environment and stakeholders involved in the design to meet user needs and expectations [28]. Co-design has become a significant element in service design projects to enable their holistic and person-centred aspirations [29]. This means working with people with lived experience from the outset, from identifying a problem to defining a solution [30] that can promote the empowerment and wellbeing of people living with dementia [31].

### 1.4. Study Purpose

The current study reports on the findings from interviews and focus groups conducted in Greater Manchester with people with mild to moderate dementia, formal and informal care partners and stakeholders from dementia, activity and volunteer services about the needs, preferences, organisational barriers and facilitators regarding participation in meaningful activities for people living with mild to moderate dementia. The aim is to map people’s needs and preferences in terms of existing community activities and the gaps and opportunities for development to identify starting points for improving or complementing existing services to promote people’s access to meaningful activities and social engagement to promote health- and wellbeing-related benefits and quality of life [14]. 

## 2. Materials and Methods

### 2.1. Research Design and Participants

To ensure the data richness and an overall perspective of these questions, this study combines the view of people with the lived experience of dementia and those with professional expertise in dementia. Focus groups and semi-structured interviews were carried out with various stakeholders from Greater Manchester between February and August 2021. Due to Covid-19 restrictions, data collection was conducted mostly via video call. Only four interviews were conducted face-to-face, as originally intended. Eleven staff members, trustees or volunteers from organisations and charities supporting the social participation and empowerment of people living with dementia (e.g., NHS, Age UK local entities, peer-support groups, and community activities) took part in one of the three focus groups. Ten of these eleven focus group participants agreed to take part in subsequent individual interviews. One participant was substituted by a colleague for schedule reasons. 

In addition, five people (three females and two males, from 58 to 78 years old) with mild to moderate dementia took part in the interviews, as well as two formal and two informal care partners. All five participants have experienced the first symptoms of the disease for at least 2 years and have a white British ethnicity background. Participants with lived experience were first approached through local contacts in dementia organisations, self-support groups and other local stakeholders in Greater Manchester on behalf of the research team. The five participants with dementia (e.g., Alzheimer’s disease, vascular dementia, mixed dementia) had a diagnosis from a local memory clinic. They were all able to communicate in the English language, had sufficient verbal fluency for taking part in semi-structured interviews, and were presumed able to give consent in response to the information provided to them [32]. Characteristics of all the participants are displayed in Table 1.

### 2.2. Data Collection and Ethics

The three focus groups (*n* = 11 participants) and the 17 individual interviews (*n* = 20 participants) were conducted by the first and the last author. Data were recorded by means of video calls using their Manchester Met Microsoft Teams accounts to comply with data requirements for dealing with sensitive research material [33,34]. To reduce the risk of fatigue, interviews were conducted as short as possible, lasting, on average, 40 to 60 min and not exceeding 90 min. The raw data recordings were deleted after the verbatim was transcribed and anonymised, usually 3–4 weeks after the recording and no later than three months after the end of the project. This study follows the ethical guidelines outlined in the Declaration of Helsinki and is approved by the Art and Humanities Research Ethics and Governance Committee of the Manchester Metropolitan University (reference number: 26864). Informed consent was obtained for all participants. 

Standardised semi-structured interviews and focus groups were conducted using a topic guide developed by the research team (see Table 2). Open-ended questions were divided into three main topics:(a)What activities are meaningful to people and for which reason?(b)What barriers and facilitators are encountered by people?(c)What opportunities can be identified for improving access and active participation in activities?

### 2.3. Data Analysis

Thematic analysis was undertaken to identify and describe the key features of participating in activities, as reported by the participants. Both inductive and deductive analyses were conducted on participants’ verbatim comments, following Guest, MacQueen, and Namey’s [35] applied approach to thematic analysis. Emerging themes, sub-themes and categories have been discussed by the research group. Based on these, the main researcher and first author developed the data codebook (see Table 3). This codebook has been discussed and refined with co-authors in an iterative process. The themes and sub-themes were further defined and clarified with two co-authors until a variance in the coding of within 10% was achieved. The transcripts and coding framework were entered into NVIVO 12 to assist the management of further qualitative data analysis.

Two a priori themes were used for the deductive analysis and derived from the interview guideline: “meaningful activities” and “barriers and facilitators”. Meaningful activities were divided into four sub-themes: favourite activities, benefits, tailored activities and involvement. Barriers and facilitators had two sub-themes: individual and environmental levels. Sub-themes and related categories are a mix of inductive and deductive approaches, with existing scientific knowledge integrated into the structure of the interview guideline; however, other sub-themes and categories emerged from focus groups and interviews and were added inductively until no new sub-themes or categories emerged (Table 3). Examples of quotes are included in Appendix A.

## 3. Results

This section reports on the qualitative insights about the needs, wishes, preferences, perceived barriers as well as self-reported strategies of people living with mild to moderate dementia regarding their involvement in sociocultural and leisure activities in Greater Manchester. The section structure follows the main themes and sub-themes of the analysis.

### 3.1. Meaningful Activities

Insights into meaningful activities are reported under the four sub-themes: favourite activities, benefits, tailored activities and involvement. 

#### 3.1.1. Favourite Activities

A large variety of favourite activities was reported by the participants, as shown in Table 4. While activities potentially relate to several categories due to their multi-dimensional nature, in Table 4, they have been linked only to their main category to aid clarity and conciseness. 

#### 3.1.2. Benefits

Benefits reported by interviewees about participation in activities have been grouped into seven categories, as shown in Table 3.

*Continuity and adaptation*: Continuity through the dementia journey is a challenge as skills and potential interests will change over the course of the progression. Continuity is perceived as a way to support people in dealing with cognitive impairment. Continuity is not doing the same activities but how to adapt to changes. At the same time, an appetite for new challenges was mentioned by people highlighting that looking for challenges is important to them and is not changed by having mild dementia symptoms. 

*Learning something new*: Participants highlighted that, despite a common stereotype that people with dementia cannot learn new things, they are often looking for challenges and developing new skills. This inclination may be expressed by the people themselves, who are actively looking for ways to learn new things (e.g., reading, meeting new people) or be nurtured by family and friends as well as formal organisations through the offering of relevant and stimulating activities. 

*Empowerment, confidence and pride*: The importance of empowerment and pride was frequently mentioned during interviews, as people living with dementia may often lack confidence in their skills and abilities. However, some people, especially those with young-onset dementia, still appreciate some competition and challenge as a way to reassure themselves of their skills and capabilities. Staff stakeholders mentioned the need to develop activities for supporting people competing against each other in a supportive context and enabling them to be aware and proud of their abilities. Confidence, empowerment and pride can be satisfied through daily living activities too.

*Staying fit physically and intellectually*: Staying physically fit is a strong motivation for participating in activities, especially since it might help slow down the progression of dementia. Staff stakeholders noted that many people with dementia are fitter than their care partners, that they can find this frustrating because they do not get the physical activity they feel they need because their care partner cannot keep up, and that physical activities offered by local groups and organisations can, in some cases, cater for this need. Regarding intellectual activities, participants living with dementia mentioned enjoying cognitive activities, such as quizzes, because they keep the mind active. Another participant explained how his wish to hike again in Scotland would provide him with some beneficial physical and cognitive stimulation by bringing old memories back. 

*Social contacts and peer support*: Participants with dementia highlighted that group activities provide opportunities to be with other people, chatting and emotionally supporting each other. Staff stakeholders mentioned that social support coming from people being or having been in the same position is also very beneficial to people, especially for those who have just received a diagnosis. It allows them to create new friendships for themselves and their care partners too, if they take part in the activities. Such peer support can be more or less formal, ranging from usual group activities to dementia cafés. People may also initiate more formalised peer support. For example, one participant, now a former carer, initiated a local dementia carers group where people living with dementia and related conditions and their care partners could seek support in a friendly environment. 

*Feeling useful to others*: The benefit of feeling useful was regularly mentioned by participants, especially in relation to offering peer support and participating in research to help future generations. It offsets the fact that people living with dementia need progressively more support and can help them feel empowered. Sharing experiences was also mentioned by familial care partners as beneficial for coping with their carer status. 

*Something other than dementia to think about:* Activities that offered something else to think about were also reported as very beneficial. The daily interactions of people with dementia and their relatives tend to focus strongly on dementia, and meaningful activities can offer new stimulus and “respite” from the challenges of dementia or of being a carer. They felt that participating together in activities helped regenerate their relationship. Participating in the same activities but having time on their own was also reported as beneficial. It provides some independence for both while participating in an activity together so that both parties can, for example, feel safe and less worried about what could happen; it can make things easier (e.g., less travel, less time spent waiting with nothing to do). 

#### 3.1.3. Tailored Activities

The data indicate that tailoring the activities is a key mechanism for improving access and active participation in activities. The following four categories emerged from the analysis. 

*Need for individualisation and personalisation*: Participants highlighted that individualisation and working with people to understand what they are looking for to enable the offering of tailored activities is key. Preferences are influenced by many factors, including age, gender, cultural background, socioeconomic groups, the range of life experiences and life trajectories. 

*Minorities’ specific preferences*: Tailoring activities should encompass the potential preferences and specificities of minority groups, e.g., ethnic and LGBTQ+ communities, and people living on their own and/or without familial support. Staff stakeholders highlighted the need for more culturally appropriate sessions or places where people can go. They reported that making activities attractive for people from minorities can be challenging, especially for those from ethnic minorities. This barrier can be reduced by providing dementia-related information or activities through people from those communities. 

*Being aware of early-onset dementia and early-stage dementia*: The need to develop more activities tailored for people with early-onset dementia was also reported, as dementia-related activities often do not meet the needs of younger people or people in the early-to-moderate stages of dementia. People in the early stages often need dedicated groups for age-appropriate physical and social activities, such as day trips or social events. 

*Activities inclusive for all*: Participants highlighted the importance of dementia-friendly facilities and groups and making activities more inclusive for everyone. They also highlighted that activities do not have to be dementia-specific. They pointed out that activities that are accessible for people with dementia will also be inclusive for a wide range of people. However, the activities must not be too easy or they may be perceived as demeaning by people with young-onset dementia or mild dementia, who are often not keen on taking part in dementia-labelled activities. 

#### 3.1.4. Involvement

Three categories linked to involvement in activities emerged from the focus groups and interviews relating to people’s motivations. 

*Initiating activities*: Insights highlighted the importance of people being able to participate actively in activities and to support opportunities for them to initiate, develop or lead activities. It is also important that groups are participant-led and that they are able to decide on the activities to engage in. In return, this helps organisations to allocate time and resources effectively and reduce the risk of running unwanted activities.

*Giving choices and decision-making opportunities*: A recurring topic was being aware of the risk of overprotecting people. Staff stakeholders emphasised the need for people to be able to engage with their chosen activity at their own level. This requires providing services and activities with some flexibility or adaptable features, such as subgroups, the possibility to take breaks or trying something more challenging, etc. 

*Not being a group person*: Service providers and other organisations usually offer group activities. However, not everyone is a “group person”. Hence, some people prefer to participate in activities on their own or in natural settings, such as spending time in a local park or shopping centre. Some people living with dementia highlighted that depending on their mood, they might prefer activities with other people or doing things on their own. In addition, a person may prefer doing an activity on their own if their skills do not match other participants’ skills. Offering opportunities to people not keen on participating in group activities can be challenging as they do not tend to be in touch with services and organisations. This makes it difficult to reach them and to understand their potential needs and how to support them. 

### 3.2. Barriers and Facilitators

The second main theme concerns the barriers and facilitators to access and participation in activities. A key issue is that the same characteristic, under different circumstances, can function either as a barrier or as a facilitator. For example, good digital literacy facilitates participation in activities, whereas low digital literacy might be a barrier, e.g., not being able to access online activity programmes or attend video calls. Characteristics mentioned by participants have been grouped in Table 5 into individual or environmental factors. 

#### 3.2.1. Individual-Level Factors

Seven categories have been identified as barriers and facilitators at the individual level. 

*Having a diagnosis of dementia*: The news of getting a diagnosis can impede participation in activities as it can take time and support for people to absorb the shock of this disclosure. Nevertheless, staff working in relevant organisations feel that the diagnosis is beneficial because more people will contact them after a diagnosis, and as a result, they can inform and support people better in accessing social and leisure activities. 

*Dementia stages*: Access and participation in activities are also impacted by the stage of dementia. Dementia-related activities are often tailored to people with mid- to late-stage dementia. For those with mild or early-onset dementia, these services appear not to be suitable (yet), but traditional community-based activities are not accessible enough either. For example, sharing activities with people with more advanced dementia is upsetting for people at the earlier stages. Similarly, people with young-onset dementia were reported as looking for more physically challenging activities (e.g., longer and faster walks, trips over several days) or for a different generational activity content (e.g., songs or content of reminiscence activities from their era). 

*Psychological factors*: Various psychological effects were reported as impacting participation in activities, such as anxiety, loss of confidence or motivation, difficulty accepting external support or help, or not feeling comfortable in group settings. Feeling self-conscious about having dementia and other people’s perceptions and dementia-denial (or anosognosia) were also reported to reduce participation. By contrast, openness to trying new activities, looking for new challenges and accepting failures as a potential outcome of taking risks helped to increase interest and engagement in activities despite the diagnosis. 

*Physical and sensorial factors*: Age- and dementia-related physical and sensorial changes can also have an impact on participation, especially the loss of coordination and dexterity and mobility issues. The negative impact of hearing and visual impairments was highlighted during the preponderance of online activities during Covid-19, making numerous people unable to participate or fully enjoy activities during video calls. 

*Cognitive factors*: Memory and attention-related deficits make participation in group activities more challenging. Participants reported struggling to remember previous sessions’ content or other participants’ names, making them, at times, feel embarrassed. Other aspects, such as a loss of spatial orientation or difficulties with language, can also negatively impact participation. 

*Communication issues*: Even if often linked to sensorial and cognitive dementia-related changes, communication issues appeared as a specific potential barrier. Staff stakeholders mentioned specific difficulties with people not having English as their first language, as they might revert back to their mother tongue due to dementia. Reading skills are often impacted, and, where possible, visual information, such as pictures, should complement the text to make it easier for people to understand (e.g., on flyers or in leaflets). Face masks used during the Covid-19 pandemic had a negative impact on communication as they tended to distort the sound of speech and make lip reading impossible.

*Habits and expertise*: Knowledge accumulated over the years supports continuity and participation in favourite activities. Skills developed long-term were reported as generally longer lasting and more resistant to dementia-related changes, supporting participation in cognitively challenging activities (e.g., music playing, speaking a foreign language, handicraft) or sharing knowledge with other people. 

#### 3.2.2. Environmental Level

The environmental level barriers and facilitators for participation in activities comprise ten categories: 

*Financial cost*: The cost of activities was frequently mentioned as a potential barrier to participating in activities. These costs can further increase when a care partner participates as well. For example, costs for transport were frequently mentioned, with participants highlighting initiatives to reduce them, such as free bus passes for older adults or disabled people or volunteer driver schemes. 

*Transport and proximity*: Difficulties of travelling were emphasised as people with dementia may not be fit enough to drive and might not have a care partner able to drive them regularly. Public transport is often not an alternative as it might not be available or might require complex journeys with several changes; some people are not familiar with using it, and there is a reluctance to use public transport since the Covid-19 pandemic. Some councils implement minibus services for people with special needs, but they are often not reliable or not convenient to join group activities. As a result, organisations and charities try to organise minibuses themselves to pick up participants or put on activities in various local venues to make them more accessible. Car sharing is also organised by some if they are accessing the same activities. 

*Facilities and amenities*: Age- and dementia-friendly features of facilities and amenities have a big impact on accessibility. The presence of a car park, bus lines with nearby bus stops, a place to buy something to eat or drink, benches to sit, or accessible toilets all increase the willingness to participate in activities. Other important aspects include good-quality and non-confusing flooring, lighting and signage and low noise levels. Small adjustments, such as making menus available online or sending them in advance, can make going to a restaurant more enjoyable as people can take their time to compare and choose what they would like to eat. Green neighbourhoods offering walks with safe footpaths that withstand bad weather were mentioned as attractive to being active outside. 

*Feeling safe*: Participants highlighted the impact of environmental features on their feeling of safety when participating in activities. For example, one interviewee mentioned she felt safer during swimming sessions when having a swimming instructor at the side of the pool as well as a hoist. Discussions also highlighted that the Covid-19 pandemic made some people less confident than before in participating in activities outside their homes due to the risk but also due to a change in habits as activities were profoundly altered by lockdowns and other restrictions. In addition, staff stakeholders emphasised living in a risk-averse society as a potential barrier. 

*Weather and seasons*: Autumn and wintertime, when it is dark, rainy and cold, which can make outside activities feel less safe, were mentioned as a potential barrier to activities, with a greater risk of falling, for example. During these seasons, the choice of activities appears to be more focused on indoor activities, and charities or organisations are perceived as a good resource for these. 

*Stigma and fear of judgement*: Participants acknowledged the need to make people more aware of dementia and make society more dementia-friendly, and stigma and fear of judgement were mentioned as potential barriers to social participation. Several organisations said they deliver dementia training to support communities and shops becoming more dementia-friendly and help people feel more confident visiting them. Badges and other stickers indicating dementia-friendliness can also make dementia more visible in society. Dementia being an invisible disability, some organisations have developed badges for people to wear to make others around them aware that they might need support. While it was acknowledged that some people might feel uncomfortable wearing them, seeing people living with dementia having sociocultural activities and enjoying themselves can also be a way to counteract negative stereotypes about dementia in public. 

*Formal network*: NHS services, service providers and the voluntary sector offer formal support for participation in activities. Such services offer a safe environment that can also inform people and families about dementia-related changes and how to adapt to them. People living with dementia and their care partners have praised the support that these organisations offer. However, they also highlighted occasional difficulties with the awareness of certain organisations and that some activity-related organisations are not keen to include people living with dementia. The need for a “one-stop-shop” contact point to inform and support people was highlighted. This formal network is especially important for people who have no family or informal support network. Participants highlighted the need for more consistency and continuity regarding activities and services. Some “postcode lottery” was mentioned, as well as favourite activities or groups being discontinued due to a lack of funding. 

*Informal network*: People living with dementia benefit from an informal network, including their spouses, siblings and children, and also friends and neighbours. Their informal networks especially support daily living activities and also leisure activities, such as taking a walk in the neighbourhood or going to a restaurant. It appears people prefer to be supported by a spouse over someone else, including their children, and not to be a burden. Professional stakeholders also noticed that spouses are sometimes reluctant to get in touch with formal networks and that children are often the ones to contact service providers or voluntary organisations to get support for their parents. The benefit of peer support was praised both by people from the formal and informal networks and as a major source of motivation for accessing services or activities. Peer support invokes trust in the advice given because it comes from people experiencing similar issues, who are less likely to judge them.

*Cultural and ethnic backgrounds*: The background of people living with dementia can influence participation: whereas in some communities, the dementia-related stigma can be strong; in others, dementia is considered a normal part of ageing. People can be reluctant to rely on organisations or charities if it is customary to look after their loved ones themselves or if people face culture-related barriers, such as the language or the kind of activities offered. 

*Online activities and virtual exchanges*: Due to the limitation of face-to-face activities during the Covid-19 pandemic, most activities were offered online. While participants praised the fact that this allowed some activities to continue to run with visual contact, they also highlighted that many people did not have access to the activities due to a lack of digital literacy or not feeling comfortable with online activities. Several organisations gave out tablets and trained people on how to use them, but it was not very successful as only a limited number of people used them. Staff stakeholders noticed that online activities were mainly accessed by people with a care partner helping them to get online. Care partners also tend to rely on instant messaging services to create virtual exchange groups to share information and informal support. 

## 4. Discussion

The current study investigates the needs, preferences, barriers and opportunities linked to participation in meaningful activities for people living with mild to moderate dementia in Greater Manchester, with the aim to inform the design of a new service for supporting access to meaningful activities for them. To provide a holistic picture, people living with mild to moderate dementia, care partners and various professional stakeholders were interviewed. 

The main objective was to have a clear understanding of which activities are generally considered meaningful in order to be able to design the service in line with people’s expectations. Participants enumerated a large variety of activities, including domestic and daily living, intellectual, physical, nature-based, music-related, craft and creative activities, socialisation and games, day trips and holidays, culture and religious activities and participation in research and advocacy activities (Table 4). This variety corroborates NICE’s [5] definition of meaningful activities, comprising leisure activities as well as some daily living activities. Knowing which activities are favoured by people living with mild dementia is essential to develop a suitable offer. The variety reported by participants highlights that there is no one-fits-all approach and that the range of activities offered needs to be as wide as possible to cater for everyone’s preferences, including minority groups’ potential specificities. Offering a wide range of tailored activities can help people find activities they enjoy, but it raises the question of how to make these available or accessible.

In order to design the service to be as relevant as possible, the study focused on currently unmet needs and how to address them. It appears that people living with early-onset dementia often look for groups to give them access to physical, cultural and social activities better suited for their younger age than the activities usually offered for people with dementia. While dedicated dementia groups can offer a unique opportunity for people to benefit from peer support, especially for people newly diagnosed, the discussions around dementia-labelled activities revealed that they might be perceived as unsuitable or stigmatising in some situations, especially for people with early-onset dementia, and that there is a need for services to be tailored to individual preferences. Participants pointed out that access could be promoted by making community activities more dementia-friendly to support people in continuing activities they had engaged in before their diagnosis rather than engaging suddenly in dementia-labelled activities. Dementia-friendly features of activities and related amenities are another way to make activities more accessible and inclusive to everyone, not only people living with dementia. The accessibility of facilities can be further increased by inviting people with dementia to share their experiences of visiting and testing places, such as football grounds, swimming pools or art galleries. Overcoming preconceptions or apprehensions and “giving it a go”, however, appears as essential as developing dementia-friendly facilities for supporting participation in meaningful activities, even challenging ones. Findings about the need to provide opportunities for people living with dementia to participate in activities on their own and to enjoy self-determination confirmed data from the literature [16]. The current study reveals that involvement in activities can range from passive participation (e.g., attending a theatre play) to being actively engaged (e.g., acting in a theatre play) and leading self-generated activities (e.g., organising a theatre play or group events). Promoting involvement also encompasses encouraging people to lead or co-lead activities by offering them opportunities and support to organise the activities (e.g., material and staff support), which requires identifying and asking them about the skills and interests they would be happy to share. Other people prefer more intimate or informal activities (e.g., impromptu chats with a neighbour, visiting a shopping centre), and some people switch between individual and group activities according to their mood or skills. This indicates that any future services will have to be flexible regarding the level of engagement, especially since dementia-related changes, as well as stigma and self-stigma, can make people less willing to participate in activities [8]. That is why the staff participants emphasised their effort to let people with dementia make choices about the activity itself and what level of challenge they are looking for. A challenge for the service to be designed will be to adapt to these different levels of expectation regarding engagement in activities and also to the varying progression of the disease and its symptoms.

Learning more about motivations for engaging in activities was also central to the study as the service might have to direct people to activities according to their motivations and values. The literature reports that people perceive activities as meaningful if they provide emotional, creative and/or intellectual stimulation [8,9]. Understanding in detail the perceived benefits of participation in activities can help make activities more appealing to people as we can tailor them to their needs and expectations. Staying fit both physically and intellectually was mentioned by participants, some of whom disclosed that they used activities as a way of trying to slow down the progression of dementia and of helping them to cope with its symptoms. 

Linked to this notion of being in control of one’s life, empowerment and pride were frequently mentioned during interviews. This can be explained by the fact that dementia tends to diminish self-confidence and opportunities for exercising control and success, whereas participation in activities can increase self-efficacy in people living with mild to moderate dementia [36]. As a result, participants in the present study reported being keen on challenges in more or less formal situations, such as running a marathon for the first time or “competing” for the most flowery garden with neighbours or family. This observation contrasts strongly with the general tendency to make activities easier for people to reduce the risk of facing difficulties due to task demands exceeding their own capacities. This desire for some competition or comparison with others deserves future investigation when collecting data on what type of competition people living with dementia favour and how the difficulty level can be tailored to people’s skills to keep it within an appropriate range. For example, one of the local organisations reported organising “Olympic games” every summer with various activities, such as athletics or archery trials with adapted materials (e.g., round foam tip safe arrows), but more importantly, with a podium, medals and an awards ceremony in the presence of family and friends. A similar focus was reported by van Corven et al. [10] as well as that people living with dementia are treated like children due to their diagnosis and there is stigma related to it. In contrast, in the present study, participants living with dementia reported setting up larger-scale challenges for themselves, such as learning to swim (again) or a several-days walking trip, being aware that it might not be successful but still being up for trying it. Familial and professional caregivers also mentioned setting small challenges related to activities to allow the person to enjoy pride and self-confidence by succeeding in them. Further, gaining new knowledge is often perceived as impossible for people living with dementia, whereas our participants reported that learning new things was a motivation to engage in activities and a source of satisfaction. Both (still looking for challenges and learning new things) indicate that a person’s interests and values do not necessarily change after the diagnosis [37]. Continuity can also be a motivation for participating in activities that the person has practised over a long time before their diagnosis, although these activities may need to be adapted over time to fit people’s capabilities. Finding a balance between continuity and trying new activities is complex, especially in group settings, as group activities have to please as many people as possible. 

The social value of activities was also preeminent among the reported benefits by enabling social contacts and peer support, the feeling of being useful to others and the possibility of focusing on something other than dementia. The results strengthen previous findings, suggesting that leisure and social activities benefit people living with dementia by providing feelings of pleasure and enjoyment, still learning new things, having a sense of connection and being useful to others, having a sense of autonomy and personal identity, as well as giving meaning to life [11,12,16]. They also highlight how engaging in meaningful activities supports empowerment, this being related by people living with dementia at home or in nursing homes as well as by formal and informal caregivers to having a sense of personal identity, having a sense of choice and control, having a sense of usefulness and being needed and retaining a sense of worth [10]. This demonstrates the high potential of interventions focusing on supporting engagement in meaningful activities to strengthen the sense of empowerment and wellbeing of people living with dementia. 

Designing a new service to support people living with mild to moderate dementia in accessing and actively participating in activities involves having a clear view of potential barriers and facilitators to taking part in activities. The same characteristic can be a barrier or a facilitator, depending on whether it is present or absent. Barriers and facilitators, therefore, can be seen to represent the two ends of a spectrum. For example, good physical health will support participation in activities, whereas poor health will impede participation. A total of 15 potential barriers/facilitators emerged from the thematic analysis relating to individual and environmental factors (Table 5). This confirms data from previous investigations showing that support by others (both formal and informal networks), suitable environmental support and physical resources are beneficial to participation in meaningful activities [8,11]. On the other hand, poor mobility and physical health, memory loss and worry, the loss of social contact and carer burden are potential barriers to participation in these activities while living with mild to moderate dementia [11]. The service to be designed will have to address these potential barriers and also people’s strengths and interests to increase the likelihood of them engaging in new activities. Indeed, discussions with participants stressed how much these various factors compound each other and how some people face multiple barriers to accessing activities. At the same time, the interviewees advocated the importance of a strength-based approach to living with dementia and how it can help to overcome some barriers. This is in line with a person-centred model for fostering a holistic approach that is not limited to cognitive and behavioural dementia-related symptoms but also takes into consideration the personality, values and history of people with dementia in order to focus on their wellbeing and independence [38]. This strength-based approach also fits with the assumption derived from the capabilities approach [39] that people living with dementia who can continue what they are capable of are more likely to experience higher levels of wellbeing. Hence, despite dementia-related limitations, there is still a lot that people with dementia can do and can offer [24], and this research advocates concrete action to support social participation through activities. A challenge to this will be to assess if these actions are efficient, as people living with dementia may have difficulties in self-evaluating subtle changes in their daily functioning due to cognitive changes, and their familial or professional caregivers might also not be aware of changes or be biased in their perception of changes regarding participation in activities and their potential benefits. However, combining various sources of information, such as people’s feedback, self-reported measures as well as wearable devices, appears to be a feasible way to align the care and health priorities of people living with dementia [40]. 

All in all, our findings confirm that more support for people with early-stage dementia is needed to access and actively participate in meaningful activities to further their wellbeing and empowerment. To do so requires clarifying people’s abilities and strengths on which to draw to facilitate participation and to estimate which kind of support they might need [7]. One strength of this study was the inclusion of the views of several people with dementia among various stakeholders, including different professional groups, to learn about their views on designing a real-world service. A limitation of this study is that as the service is intended to be implemented and tested locally in Greater Manchester, the interviews were focused only on this geographical area. Therefore, the present results may not generalise to the rest of the UK or to other countries. In addition, despite efforts to interview people of varied ethnic backgrounds, all five participants living with dementia were white British. Recruiting and interviewing participants was challenging because of the local Covid-19 restrictions at the time of this study. Because face-to-face activities were not permitted for several months, recruitment and most of the interviews had to be conducted online, limiting the variety and number of participants living with dementia. However, the consistency of the results obtained with the current literature as well as the fact that several participants are part of national organisations, giving them a larger view of living with dementia, beyond the Greater Manchester context, support the representativeness and potential generalisability of the findings. The present study highlights that while such support needs to reach and be tailored to people with a diagnosis, the offering should not be limiting or stigmatising. This provides a challenge, which was addressed in feedback presentations and discussions with stakeholders from the interviews and focus groups. Seeking validation through participant feedback is in line with developing innovative approaches to facilitate engagement in tailored meaningful activities while living at home or in care-home settings, which should be based on a co-design approach to ensure the designed product or service will fit people’s needs and capabilities [25,26,41].

From these discussions, the idea of developing a skills-exchange service emerged as the best potential solution for providing people with tailored one-to-one support, engaging in active participation and fulfilling their wish to be helpful to others. This concept has been developed in subsequent co-design workshops with stakeholders [42] while keeping in mind the potential barriers and facilitators reported in the present study. 

## 5. Conclusions

The above insights, collected through the individual interviews and focus groups, were the first step to co-designing a service to support people in accessing and actively participating in meaningful activities. They offer a broader understanding of the needs and expectations of people living with dementia regarding participation in activities and access to opportunities. They also confirm the perceived benefits of meaningful activities for people’s health and wellbeing as well as the potential barriers to overcome to facilitate participation. The results point to the potential to facilitate individualised opportunities for people living with mild to moderate dementia through participation in existing local activities and events. This engagement is presumed to be beneficial to health-related outcomes and fills a gap in the provision of activities and social participation. This insight led to the subsequent development of a skills-exchange service to offer people tailored one-to-one support for their personal preferences and to give them the possibility to help others by sharing their own strengths and skills. A special focus will be put on volunteering as it can offer an especially rewarding experience and satisfaction in helping others. The present study also provides important insights for researchers, service providers, policy makers and charities through essential knowledge about the services and support available in Greater Manchester, in the United Kingdom in general and also beyond to improve leisure and support provisions to people living with mild to moderate dementia.

## Figures and Tables

**Table 1 ijerph-20-05358-t001:** Participants’ characteristics.

Code	Focus Group (FG) ^1^ and/or Individual Interview (II)	Gender	Category	Comment
1	FG + II	Female	Staff stakeholder	Works for a local clinical commissioning group and council’s engagement and development for older people, people living with dementia and their carers
2	FG + II	Male	Staff stakeholder	Wife had dementia. He co-founded a peer-support group for people living with dementia and their relatives.
3	FG + II	Male	Staff stakeholder	Consultant old age psychiatrist
4	FG + II	Female	Staff stakeholder	Dementia adviser for a local memory loss/dementia advice service
5	FG + II	Female	Staff stakeholder	Senior fellow on brain health, service provision and co-production
6	FG + II	Female	Staff stakeholder	Project officer and dementia-walk leader in a local environmental organisation
7	FG	Female	Staff stakeholder	Manager in an independent charity offering support and services to older people
8	FG + II	Female	Staff stakeholder	Dementia support manager in an independent charity offering support and services to older people
9	FG + II	Female	Staff stakeholder	Public and patient involvement and engagement manager in a local academic health science and innovation system
10	FG + II	Female	Staff stakeholder	Dementia adviser for a local memory loss/dementia advice service
11	FG + II	Female	Staff stakeholder	Director of operations in a social marketing agency that developed a digital activity platform for people living with dementia and their relatives
12	II	Female	Staff stakeholder	Service manager in an independent charity offering support and services to older people
13	II	Female	Staff stakeholder	Manager in a local service and activity group for people living with mild to moderate dementia
14	II	Female	Staff stakeholder	Dementia worker in an independent charity offering support and services to older people
15	II	Female	Familial care partner	Husband is living with dementia
16	II	Female	Familial care partner	Mother is living with dementia
17 ^1^	II	Female	Person with dementia	Participant and volunteer for a local service and activity group for people living with mild to moderate dementia
18 ^2^	II	Male	Person with dementia	They regularly participate in an activity and social club for people living with mild to moderate dementia or memory loss
19 ^2^	II	Male	Person with dementia	
20 ^2^	II	Female	Person with dementia	
21 ^2^	II	Female	Person with dementia	

Legend: ^1^: She was interviewed by Participant 13, who is a staff member supervising this service. ^2^: Each of them were interviewed individually, accompanied by Participant 14, who was a staff member supervising the club.

**Table 2 ijerph-20-05358-t002:** Focus groups and interview topics guide.

1. Meaningful activities (physical, social and leisure activities)
Usual meaningful activities, at home or outside Activities benefits (psychological, physical, social, etc.) Access to and participation in activities during Covid-19 Preference for group or solo activities Looking for active participation, an organising role, and offering support
2. Difficulties with access to leisure activities and during leisure activities
Reasons why activities might not be accessible to people (cost, proximity, etc.) Impact of dementia on daily activities engagement, habits and learning new things
3. Adaptation, potential support, needs and wishes
Facilitators to meaningful activities participation Strategies to continue to participate in activities

**Table 3 ijerph-20-05358-t003:** Codebook.

Themes	Sub-Themes	Categories
Meaningful activities	Favourite activities	Craft and creative Domestic and daily living Intellectual Music-related Nature-based activities Physical Research and advocacy Social activities and games Day trips and holidays Culture and religion
	Benefits	Continuity and adaptation Something new to learn Empowerment, confidence, pride Staying fit physically and mentally Social contacts and peer support Feeling useful to others Something other than dementia to think about
	Tailored activities	Individualisation and personalisation Minorities Early-onset dementia and mild symptoms, inclusive for all
	Involvement	Initiating activities—active participation Not being a group person Having choices and making decisions
Barriers and facilitators	Individual level	Diagnosis Dementia stage Psychological factors Physical factors Sensorial factors Cognitive factors Communication issues Habits and expertise
	Environmental level	Financial cost Transport and proximity Facilities and amenities Feeling safe in an environment Weather and seasons Stigma and dementia understanding Formal network Informal network Culture, ethnicity and religion Online activities and virtual exchanges

**Table 4 ijerph-20-05358-t004:** Self-reported favourite activities.

Categories	Activities
1. Domestic and daily living	Baking, cooking, picking up/reading letters and newspapers, watching TV, picking fruits in the garden, shopping.
2. Intellectual	Reading, making crosswords and puzzles, quizzing, listening to book readings, memory and reminiscence activities, participating in University of the Third Age (U3A), listening to news and educational programmes, book clubs.
3. Physical	Walking in the neighbourhood, indoor walking, curling, fitness (including chair exercises), walking football, cycling (including using disability bikes), swimming, archery, hula-hoop, running, basketball (including with a softball).
4. Nature-based	Walking in nature, listening to birds, watching nature, gardening, watering plants, creating flower arrangements, caring for an animal, dog-walking.
5. Music-related	Singing, listening to music, dancing, quizzes related to music.
6. Craft and creative	Writing, poetry and playwriting, photography, drawing, painting, colouring, crafting, knitting, modelling, woodworking.
7. Social activities and games	Having a coffee/tea/drink/lunch with other people, chatting with children/grandchildren/neighbours, participating in community/dementia/memory café/buddy clubs, playing darts/snooker/bingo, quizzes, cooking together.
8. Day trips and holidays	Day trips in the surrounding area (for a football game, a walk in nature) or longer stays in hotels.
9. Culture and religion	Cinema, theatre, cultural tradition (e.g., bonfire nights, burns nights), going to the church or mosque, being a member of a congregation.
10. Research and advocacy	Medical trials/clinical research participation, public involvement events participation, being a dementia champion, organising or participating in events to raise awareness about dementia (including public speaking and conferences).

**Table 5 ijerph-20-05358-t005:** Barriers and facilitators to taking part in activities.

Potential Barriers and Facilitators	
Individual level	Environmental level
Diagnosis Dementia stage Psychological factors Physical factors Sensorial factors Cognitive factors Communication issues Habits and expertise	Financial cost Transports and proximity Facilities and amenities Feeling in a safe environment Weather and seasons Stigma and dementia understanding Formal network Informal network Culture, ethnicity, and religion Online activities and virtual exchanges

## Data Availability

The data presented in this study are available on request from the corresponding author.

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
