# Peer review of "An Investigation of the Wishes, Needs, Opportunities and Challenges of Accessing Meaningful Activities for People Living with Mild to Moderate Dementia"

_ijerph, 2023, doi:10.3390/ijerph20075358_

Round 1
Reviewer 1 Report
In this manuscript, the authors have tried to identify and map people's needs and preferences regarding existing community activities, gaps, and development opportunities. This is an important topic, and the findings may facilitate improving the existing framework to promote the social integration of individuals with dementia. I have the following comments which authors should address:
1. Authors highlighted independence in daily activity should be an essential aspect of any dementia care model. Digital health technologies are increasingly used to support dementia caregiving and promote care coordination. These technologies have the potential to improve the quality of care for people living with dementia as well as their caregivers. Authors may discuss the following study in the introduction and discussion sections:
§ Luscombe, N., Morgan-Trimmer, S., Savage, S. et al. Digital technologies to support people living with dementia in the care home setting to engage in meaningful occupations: protocol for a scoping review. Syst Rev 10, 179 (2021). https://doi.org/10.1186/s13643-021-01715-4
2. Authors identified personal preferences and goals are important aspects of determining meaningful tasks. However, it is unclear how these goals, such as walking a dog, can be tracked. I would suggest authors discuss the following study in the discussion section to highlight the potential of digital health technologies to track these goals to provide feedback to the care providers and individuals with dementia to promote shared decision-making.
§ Freytag, Jennifer, et al. "Using Wearable Sensors to Measure Goal Achievement in Older Veterans with Dementia." Sensors 22.24 (2022): 9923.
3. Authors found that young-onset dementia still appreciates some competition and challenge. This is an important finding, and the authors should elaborate further on what type of competition in the context of dementia will be appropriate. How can the difficulty level of this competition be determined?
4. Authors should provide demographic information of the individuals with dementia, including age, gender, MoCA score, education, ethnicity, and disease duration.
5. Page 2 Paragraph 1: Typo "meaning" should be replaced by "meaningful."
6. Table 2 improves the alignment of the text.
Author Response
In this manuscript, the authors have tried to identify and map people's needs and preferences regarding existing community activities, gaps, and development opportunities. This is an important topic, and the findings may facilitate improving the existing framework to promote the social integration of individuals with dementia. I have the following comments which authors should address:
- Authors highlighted independence in daily activity should be an essential aspect of any dementia care model. Digital health technologies are increasingly used to support dementia caregiving and promote care coordination. These technologies have the potential to improve the quality of care for people living with dementia as well as their caregivers. Authors may discuss the following study in the introduction and discussion sections: Luscombe, N., Morgan-Trimmer, S., Savage, S. et al. Digital technologies to support people living with dementia in the care home setting to engage in meaningful occupations: protocol for a scoping review. Syst Rev 10, 179 (2021). https://doi.org/10.1186/s13643-021-01715-4
Thank you for this suggestion. Luscombe et al. (2021) as well as Goodall, G., Taraldsen, K., & Serrano, J. A. (2021), doi:10.1177/1471301220928168, are now mentioned in the introduction and discussion sessions.
- Authors identified personal preferences and goals are important aspects of determining meaningful tasks. However, it is unclear how these goals, such as walking a dog, can be tracked. I would suggest authors discuss the following study in the discussion section to highlight the potential of digital health technologies to track these goals to provide feedback to the care providers and individuals with dementia to promote shared decision-making: Freytag, Jennifer, et al. "Using Wearable Sensors to Measure Goal Achievement in Older Veterans with Dementia." Sensors 22.24 (2022): 9923.
This suggestion has been added to the discussion section to discuss the potential of wearable devices, combined with other sources of data, to track health-related goals.
- Authors found that young-onset dementia still appreciates some competition and challenge. This is an important finding, and the authors should elaborate further on what type of competition in the context of dementia will be appropriate. How can the difficulty level of this competition be determined?
Thank you, it is indeed an important finding. We have now elaborated this in the discussion section and emphasized that future investigations would be interesting.
- Authors should provide demographic information of the individuals with dementia, including age, gender, MoCA score, education, ethnicity, and disease duration.
More demographic information has been added in the methods part as well as at the end of the discussion (p. 17) to explain that, because of the Covid-19 restrictions in place at the time of this study, our final sample was smaller and less varied than expected despite our efforts.
- Page 2 Paragraph 1: Typo "meaning" should be replaced by "meaningful."
Thank you, this typo has been corrected
- Table 2 improves the alignment of the text.
The text in table 2 has been aligned to make it more readable
Reviewer 2 Report
International Journal of Environmental Research and Public Health
Article
An investigation of the wishes, needs, opportunities and challenges of accessing meaningful activities for people living with mild to moderate dementia
REVIEW
Thanks for letting me review this interesting manuscript. The topic is relevant for clinical practice and potentially able to improve the quality of life of patients with dementia. What follows are some suggestions to improve the manuscript.
GENERAL
Some references need to be updated. Whenever possible, they should be no more than 5 years old.
The manuscript should be synthesized because it is very long and dispersive. This implies more difficulty for the reader.
INTRODUCTION
The rationale for conducting the study should emerge more clearly; for example, why there is a need to conduct an exploration of meaningful activities? What do we know so far about these activities for these patients?
Is there evidence that if patients practice more meaningful activities, their outcomes improve? Especially this latest consideration gives more strengths to the rationale for conducting the study. Niedderer defines that feelings of pleasure and enjoyment lead to well-being. But is this an opinion of an expert or there is sufficient evidence for this statement?
The fact that meaningful activities are an unmet need for these patients is important; as previously underlined, the authors should emphasize which outcomes they are supposed to improve (e.g., depression, anxiety, wellbeing, etc.).
The study purpose should be briefer and more concise and redundant parts are to be transferred in the methods (e.g., this study combines the view of people with lived experience of dementia and those with professional expertise in dementia).
RESULTS
Some reports of this section sound like as Discussion parts (e.g., Staff stakeholders noted that many people with dementia are fitter than their care partners and can find it frustrating because they do not get the physical activity they crave because their care partner cannot keep up. Therefore, physical activities offered by local groups and organisations can in some cases cater for this need). The underlined sentence seems to be part of a comment for discussion.
I suggest keeping all these comments for the discussion and limiting to reporting the results only.
The results need to be shortened and made more concise.
DISCUSSION
The first sentence needs a reference. However, this part better fits the end of the introduction, where the authors highlight the gap in the literature.
The discussion should not make references to Tables (e.g., Table 5) because this is what is to be done in the results section. Therefore, Table 5 should be presented and commented on in the results section.
Again, also this part needs to be shortened and synthesized.
I wish all authors well on their work!
Author Response
REVIEW
Thanks for letting me review this interesting manuscript. The topic is relevant for clinical practice and potentially able to improve the quality of life of patients with dementia. What follows are some suggestions to improve the manuscript.
GENERAL
- Some references need to be updated. Whenever possible, they should be no more than 5 years old.
References that are more recent have been added in the introduction and discussion sections: Chatterjee et al. (2018), Freytag et al. (2022), Goodall et al. (2021), Luscombe et al. (2021), Ortega et al. (2019)
- The manuscript should be synthesized because it is very long and dispersive. This implies more difficulty for the reader
We have synthesized the manuscript where possible to make it more concise.
INTRODUCTION
- The rationale for conducting the study should emerge more clearly; for example, why there is a need to conduct an exploration of meaningful activities? What do we know so far about these activities for these patients?
- Is there evidence that if patients practice more meaningful activities, their outcomes improve? Especially this latest consideration gives more strengths to the rationale for conducting the study.
Thank you for this comment. Interventions specifically targeting engagement in meaningful activities for people with dementia are still scarce, but first investigations are promising. This aspect is now mentioned in the introduction part. We also have added further references to evidence this point, such as the wellbeing benefits of social prescribing.
- Niedderer defines that feelings of pleasure and enjoyment lead to well-being. But is this an opinion of an expert or there is sufficient evidence for this statement?
This appears to be a misunderstanding. Niedderer does not say that feelings of pleasure and enjoyment lead to well-being. We have changed the sentence to avoid this misunderstanding.
- The fact that meaningful activities are an unmet need for these patients is important; as previously underlined, the authors should emphasize which outcomes they are supposed to improve (e.g., depression, anxiety, wellbeing, etc.).
Thank you, this aspect is now clearly mentioned in the study purpose and conclusion sections with benefits for quality of life and stress reduction (Chatterjee et al., 2018; Orgeta et al., 2019, Orgeta et al., 2022).
- The study purpose should be briefer and more concise and redundant parts are to be transferred in the methods (e.g., this study combines the view of people with lived experience of dementia and those with professional expertise in dementia).
We have addressed this point as suggested.
RESULTS
- Some reports of this section sound like as Discussion parts (e.g., Staff stakeholders noted that many people with dementia are fitter than their care partners and can find it frustrating because they do not get the physical activity they crave because their care partner cannot keep up. Therefore, physical activities offered by local groups and organisations can in some cases cater for this need). The underlined sentence seems to be part of a comment for discussion.
This sentence reports on the results, and we have rephrased it slightly to make clear that this was said by the stakeholders.
- I suggest keeping all these comments for the discussion and limiting to reporting the results only.
- The results need to be shortened and made more concise.
We have endeavoured to move any commentary to the discussion, to remove any repetitions, and to make the results section more concise overall.
DISCUSSION
- The first sentence needs a reference. However, this part better fits the end of the introduction, where the authors highlight the gap in the literature.
We have moved the sentence into the introduction and added a reference
The discussion should not make references to Tables (e.g., Table 5) because this is what is to be done in the results section. Therefore, Table 5 should be presented and commented on in the results section.
We have moved Table 5 into the results section.
- Again, also this part needs to be shortened and synthesized.
We have synthesised/shortened the section as much as possible.